# Cardiac Troponin I Antibodies Induce Cardiomyocyte Damage and Alter Cell Morphology

**DOI:** 10.3390/ijms262010005

**Published:** 2025-10-14

**Authors:** Jennifer Furkel, Vanessa A. Zirkenbach, Maximilian Knoll, Renate Öttl, Katrin Rein, Amir Abdollahi, Norbert Frey, Mathias H. Konstandin, Ziya Kaya

**Affiliations:** 1Department of Cardiology, Angiology and Pneumology, Heidelberg University Hospital, 69120 Heidelberg, Germany; 2German Center for Cardiovascular Research (DZHK), Site Heidelberg/Mannheim, 69120 Heidelberg, Germany; 3German Cancer Consortium (DKTK), German Cancer Research Center (DKFZ), 69120 Heidelberg, Germany; 4CCU Translational Radiation Oncology, National Center for Tumor Diseases (NCT), Heidelberg University Hospital (UKHD) and German Cancer Research Center (DKFZ), 69120 Heidelberg, Germany; 5Division of Molecular and Translational Radiation Oncology, Department of Radiation Oncology, Heidelberg Faculty of Medicine (MFHD) and Heidelberg University Hospital (UKHD), Heidelberg Ion-Beam Therapy Center (HIT), 69120 Heidelberg, Germany; 6Heidelberg Institute of Radiation Oncology (HIRO), National Center for Radiation Oncology (NCRO), German Cancer Research Center (DKFZ) and Heidelberg University Hospital (UKHD), 69120 Heidelberg, Germany

**Keywords:** autoantibodies, cardiomyocytes, heart failure, morphology, troponin I

## Abstract

Circulating heart-reactive autoantibodies (aAbs) detected in a variety of heart diseases (e.g., myocarditis, dilated cardiomyopathy, and myocardial infarction) have been associated with the progression of heart failure and a poor prognosis. However, the underlying mechanisms remain largely unknown. We investigated the effects of murine plasma containing aAbs against cardiac troponin I (cTnI) on neonatal rat cardiomyocytes (NRCMs). An autoimmune response to cTnI in A/J mice was induced, and anti-cTnI-aAbs were quantified. After 21 days, cardiac function, inflammation, fibrosis, and apoptosis were evaluated. In complementary in vitro liquid biopsy experiments, NRCMs were incubated with murine plasma containing high anti-cTnI-aAb levels or corresponding controls. Morphological phenotyping was performed using the C-MORE fluorescent image-based analysis workflow. Immunization with cTnI resulted in high anti-cTnI-aAb production, followed by myocardial inflammation, fibrosis, and impaired ejection fraction. NRCMs exposed to anti-cTnI-aAb-containing plasma showed reduced cell size, altered shape and radius, and elevated rate of dead cells in cell cycle analysis (*p* < 0.01, for 20% plasma). Together, these findings suggest a direct interaction of anti-cTnI-aAbs on cardiomyocytes, likely promoting adverse myocardial remodeling in vivo.

## 1. Introduction

Heart failure represents the final common pathway of numerous cardiac diseases [1]. While certain underlying causes, such as genetic predisposition, have been identified, many factors that determine and modulate disease progression remain poorly understood. Emerging evidence increasingly supports a central role of inflammatory responses in these processes. Circulating heart-reactive autoantibodies have been associated with worse prognosis and accelerated disease progression in various conditions, including myocarditis, dilated cardiomyopathy, and post-myocardial infarction remodeling. Moreover, these autoantibodies can actively contribute to disease development, as seen in autoimmune myocarditis, and may be present only in specific patient subgroups, where they influence disease progression.

To elucidate the underlying mechanisms, mouse models of experimental autoimmune diseases are commonly utilized. Immunization with cardiac proteins, such as cTnI or cardiac myosin, induces the production of high titers of anti-cTnI-aAbs, which are associated with myocardial inflammation, fibrosis, impaired left ventricular ejection fraction, and ultimately heart failure [2,3,4]. Despite these observations, the precise role of autoantibodies in disease pathogenesis remains incompletely understood. It is assumed that cardiac-specific proteins (myosin, troponin, ß-receptors, a.o.) serve as a basis for the production of autoantibodies, leading to the formation of immunological complexes that interact with cardiac structures, thereby modulating signal transduction pathways or mediating immune-mediated tissue damage [5,6,7,8,9,10,11,12]. These processes may contribute to the progressive chronification of the inflammatory burden.

Accordingly, there is evidence that anti-cTnI-aAbs possess a critical role in the development and progression of heart diseases. In this context, it has been shown that PD-1 knockout mice develop impaired myocardial function in association with the production of anti-cTnI-aAbs [5,13]. The authors demonstrated that anti-cTnI-aAbs induce chronic stimulation of calcium channels in cardiomyocytes and suggested this as one pathomechanism for the observed cardiac dysfunction [5]. Additionally, immunization with cTnI in mice can cause an immune response, leading to myocardial inflammation and impaired ejection fraction [2]. It has also been shown in clinical studies that patients without anti-cTnI-aAbs show enhanced cardiac function after acute myocardial infarction, whereas aAb-positive patients had no improvement [14]. Hence, removing aAbs from patients’ plasma can alleviate the expression of myocardial inflammation [15]. Furthermore, the development of specific cardiac antibodies shows a genetic component. Relatives of DCM patients with heart-specific and disease-specific antibodies have an increased risk of developing myocarditis and DCM [16,17]. All these findings show a correlation between heart-specific antibodies and the development of the disease. A definitive causality could not be proven so far, and thus, the role of autoimmunity is only indicated with largely unknown mechanisms. Therefore, a question arises: what role do aAbs play in this disease pattern? It was recently published that after induction of myocardial infarction in BALB/c mice, anti-cTnI-aAbs with the ability to bind membrane-type TnI on cardiomyocytes are produced. At the same time, these mice show cardiac remodeling with progressive dilatation and contractile dysfunction [18].

Taken together, heart-reactive antibodies modulate heart diseases on a very broad spectrum. They have been described as predictors for prognosis and might represent a promising target for novel therapies. Animal models for autoimmune myocarditis have been developed to study the disease. In our model of cTnI-induced experimental autoimmune myocarditis EAM, inflammation of the heart can be induced by repeatedly immunizing susceptible animals, in this case A/J mice, with either cardiac full-length protein or peptide sequences [2,19]. As a result, immune cells migrate into the myocardium and impair heart function, similar to the human disease [2,19,20]. A benefit of this model is that, in contrast to the viral model, the developed inflammation is solely attributed to autoimmune pathomechanisms. In the viral model, inflammation is induced but is also accompanied by reactions against the virus [11,21,22]. This autoimmune phase can also be observed in patients. Despite the successful elimination of a viral pathogen, immune reactions can persist, potentially resulting in autoimmune myocarditis [20,23]. Consequently, the cTnI-induced myocarditis model offers a valuable opportunity to investigate potential endogenous immune responses. Our experience shows that even within the mouse strain, inflammation can vary in severity. The spectrum of symptoms in humans also demonstrates considerable variation in severity in some cases [20,23]. They both also have the production of circulating autoreactive antibodies against the heart’s own proteins in common [2,14,19,20,23]. Plasma containing these antibodies was tested for its effect on cardiomyocytes. Morphological and cellular readouts have been utilized to recognize disease patterns in cultured cardiomyocytes. Here, we investigated the direct effect of antibodies on cardiomyocytes regarding morphological characteristics based on an autoimmune model of cTnI-induced EAM. The aim of this study was to elucidate the significance of the humoral immune response by means of autoantibodies. These morphological changes can provide insights into the vitality and properties of cardiomyocytes, contributing to a better understanding of the processes involved.

## 2. Results

### 2.1. Immunization with cTnI Leads to the Production of High Anti-cTnI-aAb Titers Followed by Myocardial Inflammation, Fibrosis, and Reduction in Ejection Fraction

It has been postulated that aAbs against cardiomyocyte surface antigens impair cardiac function [5,14]. In this study, we aimed to investigate the effects of immunization with cTnI and subsequent induction of aAbs against cTnI antigens. An overview of the experimental design and measured readouts is provided in Figure 1A.

To induce an autoimmune response against cTnI, as previously described [2], mice were immunized with murine cTnI on d0 and d7. Control groups included control buffer-treated and untreated mice. Mouse plasma collected on d21 showed high anti-cTnI-aAb titers in cTnI-treated mice, whereas in control buffer-immunized and untreated mice, no autoantibodies were detected (1:266,240 ± 142,443 vs. 0 vs. 0; ** *p* = 0.008, ** *p* = 0.008) (Figure 1B). To exclude cross-reactivities, we measured cTnT and myosin antibody titers. In comparison to the production of anti-cTnI-aAbs, only a very low titer of cTnT (1:0.848 ± 0.267 vs. 0 vs. 0; * *p* = 0.016) (Figure 1C) and myosin (1:0.116 ± 0.052 vs. 0 vs. 0; * *p* = 0.016) (Figure 1D) could be measured. Therefore, cross-reactivity does not play a role. Immunization with cTnI leads to myocardial damage, resulting in highly elevated hsTnT levels in cTnI-treated mice (range: 60–4020 pg/mL) compared to both controls (1504 ± 670.3 vs. 0 vs. 0; * *p* = 0.016, ** *p* = 0.008) (Figure 1E). The lowest value of 60 pg/mL is, nevertheless, above a pathological threshold. We calculated the heart weight to body weight ratio (HBR) on d21, which was significantly lower in cTnI-immunized mice as contrasted to control buffer-immunized and untreated mice (2.876 ± 0.357 vs. 3.925 ± 0.125 vs. 4.625 ± 0.206; * *p* = 0.041, ** *p* = 0.006) (Figure 1F). Because the body weight influences the HBR, murine body weight is shown in Figure 1G. Body weight was reduced in cTnI-treated mice compared to control buffer-treated and untreated A/Js (16.67 ± 0.597 vs. 18.0 ± 0.408 vs. 17.68 ± 0.704; *p* = 0.175, *p* = 0.730).

Next, we examined alterations in cellular composition and the extracellular matrix by calculating the inflammation score from HE staining and the fibrosis score from AFOG staining. Immunization with cTnI compared to control buffer and untreated led to an increase in inflammation (45 ± 11.73 vs. 0 vs. 0; * *p* = 0.016, ** *p* = 0.008) (Figure 2A) and fibrosis (26.2 ± 7.41 vs. 0 vs. 0; * *p* = 0.048, * *p* = 0.048) (Figure 2B). An additional automated fibrosis analysis is provided in the Appendix A. Increased collagen deposition and high cellularity, indicative of inflammatory cells, as well as corresponding macroscopic hearts, are shown in representative macroscopic images in Figure 2C,D. The macroscopic image shown for cTnI immunization was chosen because of the clearly visible inflamed areas on the heart. This heart was both the most inflamed and heaviest.

Lastly, we measured left ventricular ejection fraction with echocardiography to assess cardiac function. Here, immunization with cTnI led to a significant reduction in ejection fraction 21 days after the first immunization, whereas both control groups were unaffected (68.9 ± 3.34 vs. 93.34 ± 0.62 vs. 92.72 ± 0.63; *** *p* < 0.001, *** *p* < 0.001) (Figure 3A). Representative echocardiographic images are shown on the right in Figure 3B. No difference could be detected in the heart rate of the mice at the end of the experiment (651.43 ± 44.75 vs. 644.24 ± 39.26 vs. 664.58 ± 42.53) (Figure 3B). Additional echocardiographical parameters are provided in Appendix A.

### 2.2. Plasma from cTnI-Immunized Mice Alters NRCM Morphology and Induces Apoptotic Cell Death

To assess the direct effects of anti-cTnI-aAbs, neonatal rat cardiomyocytes (NRCMs) were incubated with 5% or 20% pooled plasma collected on day 21 from cTnI-immunized, buffer-treated, or untreated control mice. After 48 h of incubation, cellular phenotypes were analyzed using our in-house developed C-MORE workflow, an image-based high-content morphological assessment tool [24]. Immunofluorescent staining and automated microscopic imaging were performed using DAPI for nuclei, desmin–TexasRed for the cytoskeleton and sarcomeres, and an NFAT (nuclear factor of activated T cells)–GFP reporter to monitor pro-hypertrophic NFAT–calcineurin signaling. Cells were segmented from images, and >500 features were quantified per cell using CellProfiler [25]. Morphological classification and profiling were conducted using the cmoRe R package v.1.0 [24].

In the initial step, cell vitality was assessed using DAPI intensity and cell/nucleus size ratios. Cell cycle status was assigned based on DAPI intensity, and nucleus morphology was used to exclude remaining fibroblasts (~5%) from the primary cell preparations. These cmoRe quality control and filtering steps ensured that downstream analyses focused on pure, viable cardiomyocytes while also quantifying and excluding dead cells. Representative images of NRCMs incubated with plasma from control or cTnI-immunized mice are shown in Figure 4A. Morphologically, NRCMs in the control groups appeared larger and more branched (Figure 4A,B), while the total number of particles per field of view was comparable between groups.

In Figure 4C, cell size quantifications of pure cardiomyocytes are shown, after the exclusion of fibroblasts, debris, and dead cells. Morphological analysis using C-MORE demonstrated a significant reduction in cell size following incubation with 20% cTnI plasma (3714 [cTnI] vs. 7174 [cb] vs. 6877 [ut], s.e.: 697; p[cTnI vs. cb] < 0.001; p[cTnI vs. ut] < 0.001). Incubation with 5% plasma induced smaller but statistically significant differences (6281 [cTnI] vs. 5859 [cb] vs. 5712 [ut], s.e.: 293; p[cTnI vs. cb] = 0.04; p[cTnI vs. ut] = 0.005. Taken together, cTnI induces a significant decrease in mean cell size.

Analysis of cell cycle distributions revealed a significantly higher fraction of dead cells in NRCMs treated with 20% cTnI plasma, suggesting an induction of cell death by cTnI plasma (Figure 4D,E) (19.8% [cTnI] vs. 6.5% [cb] vs. 5.8% [ut], s.e.: 4.3%; p[cTnI vs. cb] < 0.001; p[cTnI vs. ut] < 0.001). No significant difference in dead cell fractions was observed at 5% plasma incubation (4.4% [cTnI] vs. 3.9% [cb] vs. 4.4% [ut], s.e.: 0.9%; p[cTnI vs. cb] = 0.63; p[cTnI vs. ut] = 1.00) (Figure 4D,E). G2/S fraction was decreased in the former (7.7% [cTnI] vs. 28.4% [cb] vs. 20.2% [ut], s.e.: 6.7%; p[cTnI vs. cb] = 0.002; p[cTnI vs. ut] = 0.06) but not in the latter group (30% [cTnI] vs. 18% [cb] vs. 19.8% [ut], s.e.: 6.3%; p[cTnI vs. cb] = 0.16; p[cTnI vs. ut] = 0.23). To further determine the type of cell death induced by cTnI treatment, we performed a TUNEL assay on histopathological heart slides, revealing a significantly higher signal in the cTnI group compared to the control buffer or untreated mouse plasma (13.96 ± 2 vs. 0 vs. 0; ns) (Figure 4F), suggesting a direct induction of apoptotic cell death.

Finally, we performed the full multiparametric phenotyping of cell morphology using the cmoRe R package. Starting from 1216 well-aggregated features, *n* = 81 differential features (5% cTnI) and *n* = 845 features (20% cTnI) were selected (Benjamini–Hochberg *p* < 0.05, linear mixed model analysis), resulting in nine (5% cTnI) and twenty-two (20% cTnI) meta-features, which are shown as radar plots in Figure 5A (mean aggregation). Visually, a high concordance between untreated controls and control buffer samples (yellow/green) can be observed, whereas cTnI treatment induces a distinct phenotype shift (Figure 5A). Top differential features are shown in Appendix A.

### 2.3. Single-Cell Morphology Analysis Identifies Two Distinct cTnI Plasma NRCM Subtypes

The previous C-MORE analysis demonstrated differences in the average morphology phenotype of cells (Figure 5A). However, such differences do not necessarily arise from a uniform alteration of single-cell phenotypes. We, therefore, aimed to further deconvolute the observed changes on the single-cell level. We computed two-dimensional representations from 1216 features/dimensions for each cell using UMAP (Figure 5B). Utilizing our previously described in silico filtering steps, we ensured to only include vital single cells in our morphological analysis.

NRCMs treated with 20% cTnI plasma displayed the largest deviation from controls (Figure 5B, upper row, left column, red arrows); NRCMs treated with 5% plasma showed minor differences compared to controls (Figure 5B, bottom row).

To further deconvolute the altered populations following 20% cTnI plasma treatment, we recalculated low-dimensional representations from only 20% cTnI plasma data. We visually combined 12 k-means clusters into 5 main clusters, suggesting the induction of NRCM subtypes in the cTnI plasma-treated NRCMs (Appendix A).

Taken together, cTnI plasma treatment led to a significantly reduced cell size in NRCMs. The cell death rate is increased in these NRCMs. Histopathological assessment could specify an increased rate of apoptosis, and single-cell morphology suggests the induction of differential NRCM subtypes.

## 3. Discussion

The role of autoimmunity, especially of cardiac-specific aAbs, on cardiomyocytes and the underlying mechanisms is still not completely understood.

In this paper, we addressed the effect of mouse plasma with high anti-cTnI-aAb titers from cTnI-immunized animals on NRCMs. First, we immunized mice with cTnI or control buffer and used untreated controls to collect plasma and the heart. In the next step, NRCMs were incubated with collected plasma at a concentration of 5% or 20% in cell culture medium. Single-cell morphology analysis was performed using C-MORE, collecting information about different phenotype features. Overall, we showed that plasma with high anti-cTnI-aAbs titers is able to change the cell morphology of cardiomyocytes compared to the plasma of the control buffer and the plasma of untreated mice.

In this study, we used an autoimmune mouse model to induce the production of anti-cTnI-aAbs by the immunization of A/J mice with cTnI. After immunization, high anti-cTnI-aAb titers could be detected in these mice, followed by significant myocardial inflammation and fibrosis. The infiltrating cells are predominantly CD4+ T cells, B cells, and CD8+ lymphocytes [26,27]. The depletion of these cells has been demonstrated to prevent myocarditis [28]. Conversely, adaptive T cell transfer has been observed to induce myocarditis in susceptible recipients [29,30]. Furthermore, cTnI-immunized mice showed a decrease in cardiac pump function on d21. Evaluating the HBR, cTnI-immunized animals showed a reduction due to a decrease in heart weight, suggesting apoptotic processes in these mice, while control buffer and untreated mice are hypertrophied with age [31]. Immunohistological examination revealed an increase in apoptotic cells and increased cardiac damage marker hsTnT. Immunological mechanisms lead to tissue-based changes called cardiac remodeling [32,33]. We hypothesize that these effects are mediated at least in part by the specific anti-cTnI-aAbs since both of our control groups show no inflammation, fibrosis, or effect on the heart function. This is supported by the findings of Okazaki et al., who demonstrated in PD-1 knockout mice that anti-cTnI-aAbs are responsible for developing spontaneously autoimmune dilated cardiomyopathy in these mice by the administration of monoclonal antibodies [5]. Furusawa et al. showed autoantibody production in BALB/c mice after myocardial infarction, accompanied by reduced pumping function of the heart and enhanced cardiac inflammation [18]. Furthermore, the administration of monoclonal autoantibodies has been observed to induce myocarditis and associated myocardial dysfunction [5,34]. In the clinical setting, the elimination of autoantibodies from the patient’s circulation has been demonstrated to exert a favorable impact on the progression of the disease [14,35,36].

We collected plasma from cTnI and control buffer-immunized mice, as well as from untreated mice, and used healthy NRCMs in cell culture to evaluate the direct effects of the pooled autoantibody-containing plasma on the cardiomyocytes. Morphological profiling showed significantly altered profiles compared to control buffer-immunized and untreated mice. Most prominently, the DAPI-based cell cycle analysis showed increased rates of cell death. This effect could further be increased by increasing the total plasma concentration in the cell culture medium (5% and 20%) on NRCMs. Interestingly, the TUNEL assay of the histopathological material of immunized mice hearts also showed increased apoptotic cells. Apoptosis is a method to cleanly remove damaged cells without interfering with surrounding cells and without inducing the endogenous immune system. Apoptotic cells characteristically show a decreased cell size caused, among other things, by the influx of Na^+^, leading to a typical apoptotic volume decrease, before the cell starts further disintegration processes [37,38,39,40]. We found a significant in vitro cell size reduction in high cTnI plasma concentration on cardiomyocytes, which seems to be a damaging effect due to the higher number of stimuli, whereas a lower concentration of plasma containing anti-cTnI-aAbs leads to an increase in cell size in the course of remodeling mechanisms to restore cellular normality. These results fit with the observation that Wu et al. made. They showed that anti-cTnI-aAbs could induce myocardial dysfunction via induction of a signaling pathway responsible for myocardial apoptosis, myocardial collagen deposition, and impaired systolic dysfunction [41]. As a result, the cardiac tissue attempts to regain a normal state through cardiac remodeling. However, the maintenance of inflammation sooner or later leads to the targeted death of cells.

Looking more closely at the density distribution of the cell population, the 20% cTnI plasma concentration reveals prominent individual NRCM subgroups, each differing in morphological properties. This suggests direct interaction between anti-cTnI-aAb-containing plasma and NRCMs, inducing cellular shape and texture remodeling in the NRCMs and leading to further subtypes.

Taken together, we could validate a functionally impaired, fibrotic, and inflamed cardiac phenotype in cTnI-immunized mice. Other studies have already shown that aAbs can induce and support disease progression. In this case, the pathogenic effect of AAbs on the course of the disease is also possible. The plasma, which contained high titers of anti-cTnI-aAbs, could possibly elicit cytoskeletal remodeling in direct interaction with NRCMs in vitro. These results suggest a potential direct interaction of anti-cTnI-aAbs and cardiomyocytes as an important mechanism of action. In the future, they could be utilized as therapeutic targets in patients with detected heart-reactive antibodies. However, further studies are needed to elucidate the role of cardiac antibodies in cardiovascular diseases.

## 4. Materials and Methods

### 4.1. Mice and Induction of Autoimmune Myocarditis

Female A/JÓla (A/J) wild-type mice (5 weeks) obtained from Envigo were maintained in the animal facility unit of the University of Heidelberg. All procedures involving the use and care of animals were reviewed and approved by the Institutional Review Board/Ethics Committee of the regional council of Karlsruhe (Animal application G-4/21; date of approval 24 March 2021) and by the Animal Care and Use Committee of the University of Heidelberg (German Animal Protein Code). The investigation was conducted in accordance with the German animal welfare act (Directive 2010/63/EU).

To induce EAM [2,19], mice [*n* = 5] were subcutaneously immunized twice below the armpit near the lymph nodes on day 0 (d0) and 7 (d7) with 100 µL emulsion containing 150 µg cTnI in complete Freund’s adjuvant (CFA), which was supplemented with 5 mg/mL mycobacterium tuberculosis H37Ra (Sigma, St. Louis, MO, USA). The emulsion was freshly prepared before each immunization to ensure its proper function. Murine cTnI was provided and prepared as already described [42,43,44]. For control [*n* = 5], 1× PBS buffer in CFA was used for immunization with control buffer (control). One group of mice [*n* = 4] received no treatment (untreated). On day 21 (d21), mice were anaesthetized by intraperitoneal injection of 120 mg/kg ketamine and 16 mg/kg xylazine, and blood was taken for plasma extraction. The experiment was terminated by cervical dislocation. Heart and other organs were explanted for further analysis. The schematic timeline for the experimental setup is shown in Figure 1A.

### 4.2. Determination of Highly Sensitive Troponin T (hsTnT) and Anti-cTnI-aAbs

Plasma from d21 was used to detect myocardial damage by hsTnT analysis and detection of anti-cTnI-aAbs by ELISA. hsTnT levels were analyzed by electrochemiluminescence (Elecsys 2010 analyzer, Roche Diagnostics, Mannheim, Germany). Plasma was diluted 1:10 with a cold 0.9% NaCl solution. Details of the test principle have been described before [45].

Anti-cTnI-aAb titers were determined using the ELISA technique. These were performed as described before [46,47]. Therefore, 96-well microtiter plates were coated with 100 µL/well of cTnI (5 µg/mL) in bicarbonate buffer (pH 9.6) and incubated overnight. Anti-mouse secondary IgG antibodies (Sigma, St. Louis, MO, USA) were diluted to 1:1000 and used for detection. Plasma samples were diluted for anti-cTnI-aAb detection to 1:800, 1:3200, 1:12,800, 1:51,200, 1:204,800, 1:819,200, 1:3,276,800, and 1:13,107,200. Color reaction was developed with 100 µL TMB-HRP and stopped by adding 100 µL of 1 M H_2_SO_4_ to each well. Optical densities were determined at 450 nm with a reference filter of 540 nm. Antibody endpoint titers for each individual mouse were calculated as the greatest positive dilution of an antibody reaction.

### 4.3. Histopathological Preparation and Manual Analysis

Explanted hearts were cut longitudinally and vertically to the septum into two pieces. One half was fixed in 10% formalin and subsequently embedded in paraffin. Six sections (2–3 µm) from different cardiac layers were cut and stained with hematoxylin and eosin (HE) to determine the level of inflammation or with acid fuchsin orange G-stain (AFOG) to investigate fibrosis. Staining was performed according to standard protocol using standard reagents. All sections of each heart were inspected in a double-blind manner by two independent investigators who were blinded to the treatment and immunization status of the animals and groups. Therefore, under the light microscope, the affected area of infiltration or fibrosis was considered in relation to the whole heart section by eye and was specified as a percentage. The mean values of inflammation or fibrosis were then calculated from the values of both investigators and indicated as a percentage.

### 4.4. Automated Fibrosis Area Quantification on Histopathological Sections

To quantify fibrosis, the percentage of fibrotic tissue in the entire myocardium was measured. For this purpose, blue-stained areas were visualized using a color threshold, and the images were converted to grayscale. In addition, the same images without a threshold were converted to grayscale to calculate the total tissue area. The images were then analyzed without including perivascular fibrosis or intramural vascular structures [48,49].

### 4.5. Echocardiography

Echocardiographic measurements were performed without anesthetizing the mice on day −1 (one day before immunization start) and day 20 using the Visual Sonics Vevo 2100 system 30 MHz linear MicroScantransducer (MSH400, VisualSonics, Toronto, ON, Canada), specially optimized for cardiovascular experiments in mice. Parasternal long-axis projection cine loops were acquired at the level of clear visible walls of the aortic annulus. Ejection fraction and fractional shortening were determined by appropriate software, provided with the Vevo2100 platform.

### 4.6. NRCM Isolation and Treatment

NRCMs were obtained from hearts of 1–2-day-old neonatal rats using a trypsin-based enzymatic digestion standard protocol as previously described [24]. To enrich cardiomyocytes, a Percoll gradient centrifugation was performed after digestion. Cells were then counted and plated onto 96-well plates optimized for imaging (Corning, #353219, Corning, NY, USA). The plates were coated with laminin (Sigma, St. Louis, MO USA; L2020, diluted to 1 mg/L PBS) for 2 h beforehand, and 20.000 cells per well were seeded in 200 μL high-plasma medium (10% FBS in M199). After 24 h, the cells were incubated with the NFAT-GFP adenovirus, following the C-MORE protocol as previously reported [24]. Then, the medium was changed, and the cells were incubated for 48 h with media containing 5% and 20% pooled mouse plasma from cTnI-immunized mice or control with low FBS plasma.

### 4.7. NRCM Immunocytochemistry

In all NRCM experiments, cells were fixed with 4% paraformaldehyde (PFA), permeabilized with 0.2% Triton X-100, and then incubated in blocking solution (10% FBS in PBS) for one hour at room temperature. In all NRCM experiments, cytoskeletal staining using a desmin antibody (1:800) was applied on the plates overnight at 4 °C. After washing with PBS, the secondary antibody (1:600) was applied for 1 h. Nuclear DNA was stained in all experiments using 4′,6-Diamidin-2-phenylindol (DAPI).

### 4.8. NRCM Image Acquisition (IN Cell Analyzer 2200)

Images were acquired in an automated fashion using the IN Cell Analyzer 2200 (GE Healthcare Europe, Freiburg, Germany). For all NRCM experiments, a centered square arrangement of a 4 × 4 grid of imaging fields per well was chosen, and three channels were imaged (DNA—DAPI, desmin—TexasRed, NFAT—GFP native fluorescence/FITC). Images had a resolution of 2048 × 2048 pixels.

### 4.9. Data Processing

Cellular morphology features were calculated and extracted from the images using the freely available CellProfiler software version 3.1.8. Cell and feature processing was performed using functions of the custom-written R package cmoRe, publicly available on GitHub (https://github.com/mknoll/cmoRe, accessed on 31 October 2024). In short, features reflecting the treatment effect were normalized and aggregated to meta-features, improving the robustness of the results. Appendix A demonstrate which features were used to construct the meta-features. Appendix A further demonstrates the reproducibility of the results across biological replicates. The NFAT score was calculated by automated thresholding on the median GFP intensity of the nucleus. It reflects the translocation of NFAT-GFP from the cytoplasm compartment into the nuclear compartment upon activation of the calcineurin–NFAT pathway. Therefore, the presence of NFAT-GFP in the nucleus indicates activation of the pro-hypertrophic NFAT–calcineurin pathway. For cell cycle analysis, we used the integrated DAPI intensity of the nucleus; for the identification of non-attached cells, we used the cellular/nuclear area ratio; and for the identification of non-cardiomyocytes, we used the median DAPI intensity of the nucleus.

### 4.10. Single-Cell Phenotyping

Filtered, non-aggregated data of the highest concentrations of tested substances were analyzed with t-SNE using the FI-tSNE implementation, and Uniform Manifold Approximation and Projection (UMAP) similarity between substance-induced phenotypes was assessed by binning the two-dimensional t-SNE data (*n* = 50 bins), normalizing data (rates), and calculating pairwise differences. The 5% and 95% quantiles of the combined unique pairwise combination differences were used to indicate significance.

### 4.11. Statistical Analysis

Statistical analysis for data from in vivo experiments was performed using GraphPad Prism version 7.00 for Windows (GraphPad Software, La Jolla, CA, USA). All data are plotted as individual points. Data summaries are given as mean ± SEM. Normal distribution of the control group was tested using the D’Agostino–Pearson normality test. All tests used were two-tailed. For group comparison with repeated measurements, one-way analysis of variance was performed, followed by a multiple comparison test. The threshold of significance for all tests was set at 0.05. Statistical analyses for in vitro morphology readout were performed utilizing linear models as previously described within the cmoRe assay workflow [27].

## 5. Conclusions

Heart-reactive antibodies are present in several heart diseases and are considered as predictors for prognosis, and, therefore, they are a promising target for novel therapies. Immunization of A/J mice with cTnI induces the production of anti-cTnI-aAbs and the development of myocardial inflammation. This murine plasma containing anti-cTnI-aAbs has the ability to alter the morphology of NRCMs, induce apoptotic cell death, and generate two distinct cTnI plasma NRCM subtypes.

## Figures and Tables

**Figure 1 ijms-26-10005-f001:**
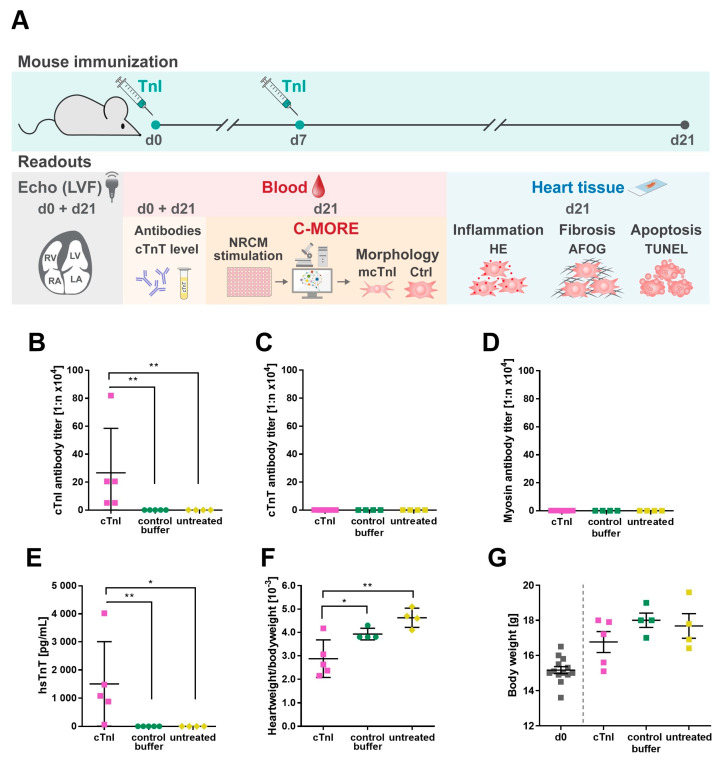
Induction of EAM by immunization of A/J mice with cTnI compared to control groups. Mice were immunized with 150 µg cTnI (*n* = 5) or control buffer (*n* = 5) on day 0 and 7 and sacrificed on day 21. Untreated mice (*n* = 4) were used as an additional control. (**A**) Schematic overview of the experimental in vivo and in vitro setup. Examination of anti-cTnI-aAb (**B**), anti-cTnT-aAbs (**C**), and myosin-aAb titers (**D**) of mouse plasma collected on d21. (**E**) Determination of hsTnT levels in the plasma of d21. Calculated ratio of heart weight to body weight (**F**) and measured body weight (**G**) on d0 and d21. Data are displayed as mean ± SEM. Statistical analysis was performed using two-way ANOVA with the Bonferroni post hoc test and the Mann–Whitney test for skewed data. *p* values of *p* < 0.05 were considered statistically significant and marked by * *p* < 0.05 and ** *p* < 0.01.

**Figure 2 ijms-26-10005-f002:**
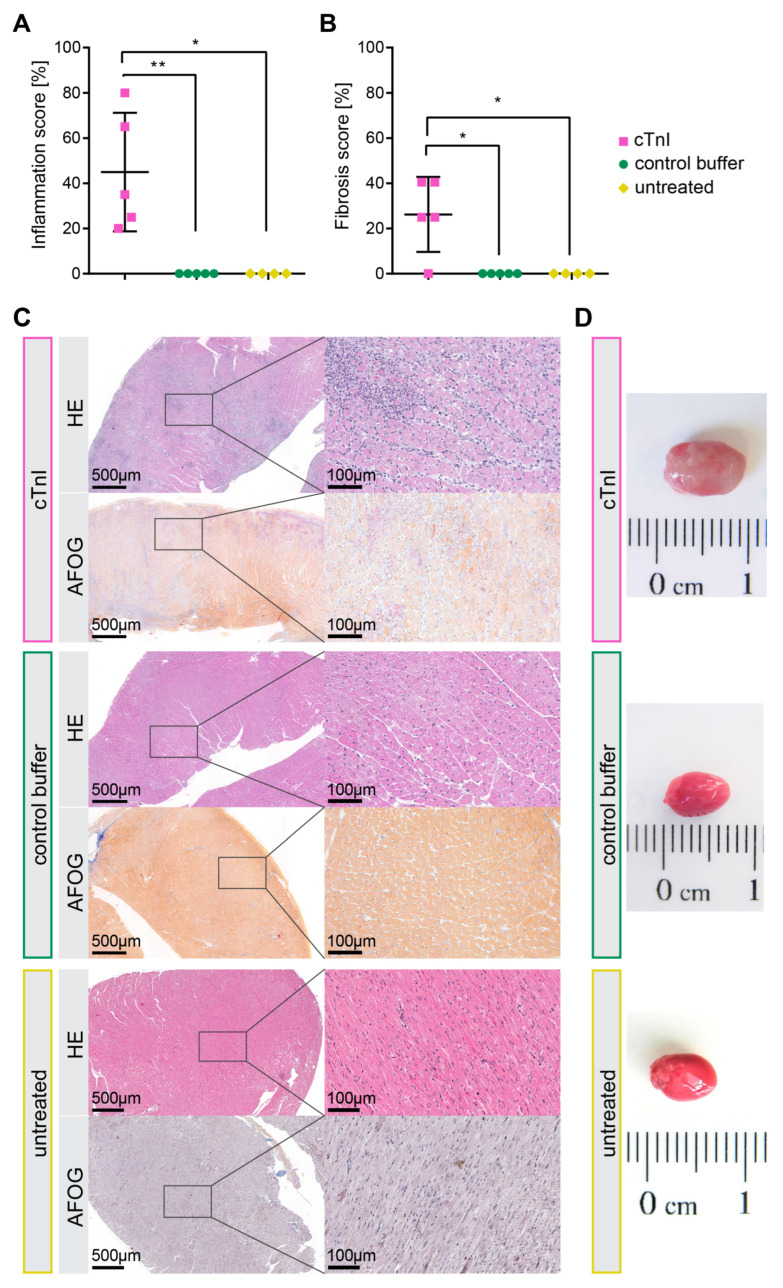
Evaluation of tissue alteration in the myocardium of cTnI (*n* = 5) or control buffer-treated (*n* = 5) and untreated (*n* = 4) A/J mice. Paraffin-embedded heart tissues were stained using HE or AFOG. Areas of infiltrated immune cells (**A**) or fibrosis (**B**) were considered in relation to the whole heart section and were specified as a percentage. The means in percentage were calculated from the values of two experienced readers. (**C**) Representative pictures of murine heart sections with inflamed and fibrotic areas are in magnifications of 4-fold and 20-fold, as well as a macroscopic example heart (**D**). Statistical analysis was performed using the Mann–Whitney test. *p* values of *p* < 0.05 were considered statistically significant and marked by * *p* < 0.05 and ** *p* < 0.01.

**Figure 3 ijms-26-10005-f003:**
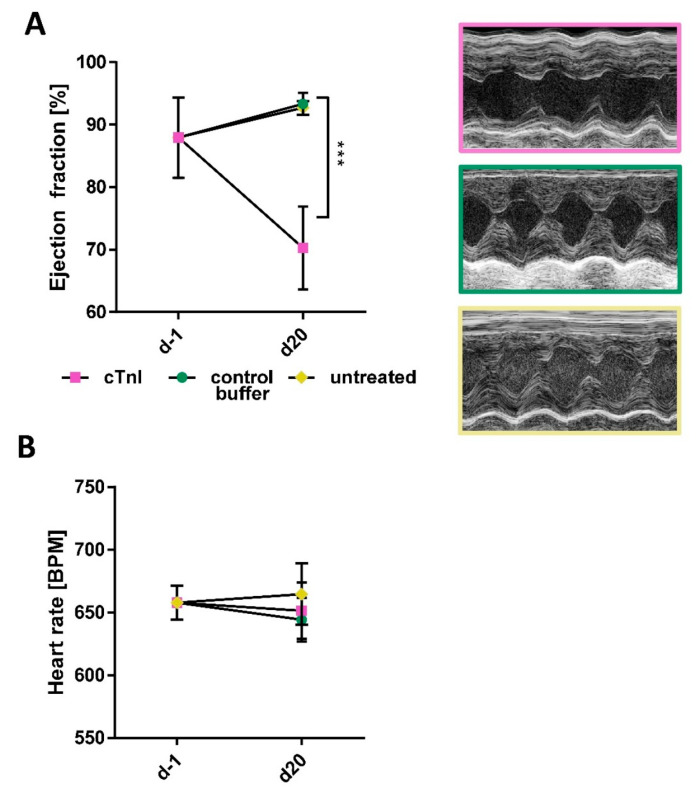
Assessment of cardiac function in cTnI (*n* = 5) or control buffer-treated (*n* = 4) and untreated (*n* = 4) A/J mice. (**A**) Echocardiographic analysis of ejection fraction on d21 was analyzed using M-mode. Statistical analysis was performed using two-way ANOVA with the Bonferroni post hoc test. *p* values of *p* < 0.05 were considered statistically significant and marked by *** *p* < 0.001. Representative M-mode echocardiographic images are shown for cTnI, control buffer, and untreated mouse groups. (**B**) Heart rate of A/J mice during echocardiographic assessment.

**Figure 4 ijms-26-10005-f004:**
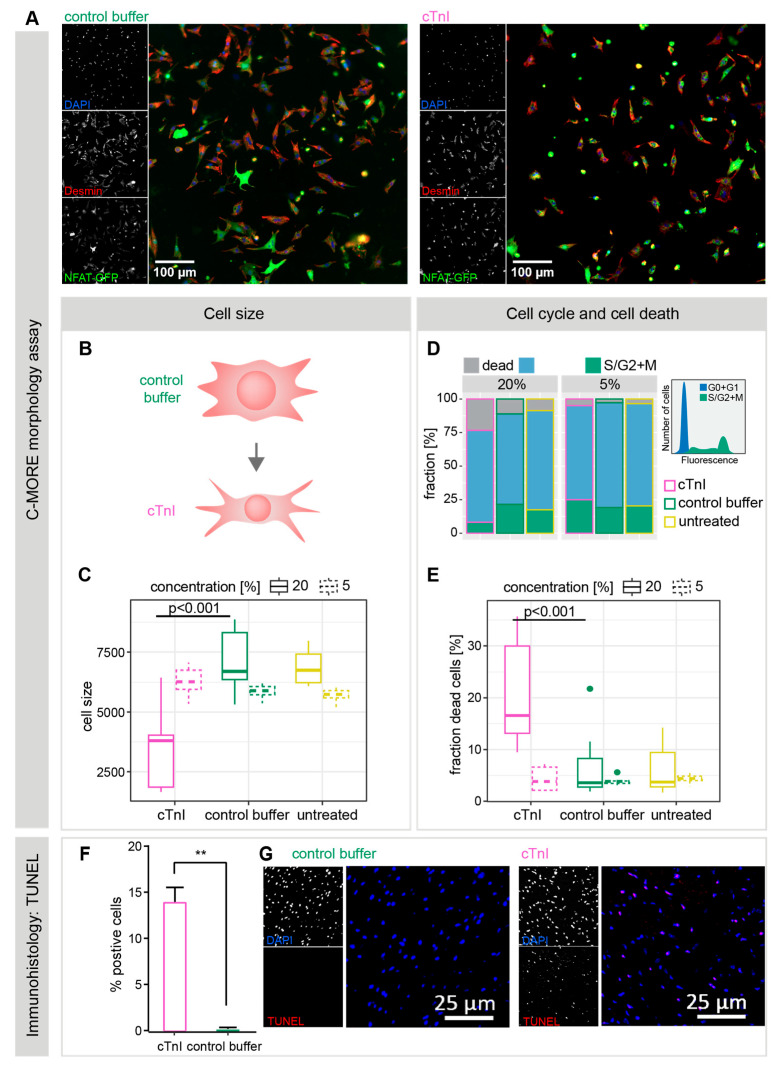
Plasma of cTnI-immunized mice leads to morphological changes of cardiomyocytes in vitro. (**A**) Exemplary fluorescence microscopic images of NRCMs incubated with control buffer (**left**) and cTnI-immunized mouse plasma (**right**). Cell nuclei are stained with DAPI, cytoplasma with desmin and reporter virus with NFAT-GFP. (**B**) Illustration of morphological phenotypes of control buffer and cTnI plasma-treated NRCMs. (**C**) Evaluation of in vitro cell size of NRCMs after incubation with 5% and 20% plasma concentration of cTnI or control buffer-treated and untreated mouse plasma. (**D**) In vitro cell cycle analysis of cells stimulated with 5% and 20% plasma. (**E**) Fraction of dead cells in percent calculated with the DAPI fluorescence-based cell cycle tool of cmoRe in vitro. (**F**) TUNEL assay in heart sections of mice immunized with cTnI and control buffer. Data are displayed as mean ± SEM. Statistical analysis of in vivo data (cell size, TUNEL) was performed using the Mann–Whitney test for skewed data. *p* values of *p* < 0.05 were considered statistically significant and marked by ** *p* < 0.01. (**G**) Representative immunohistology images of the TUNEL assay in heart sections of mice immunized with cTnI and control buffer.

**Figure 5 ijms-26-10005-f005:**
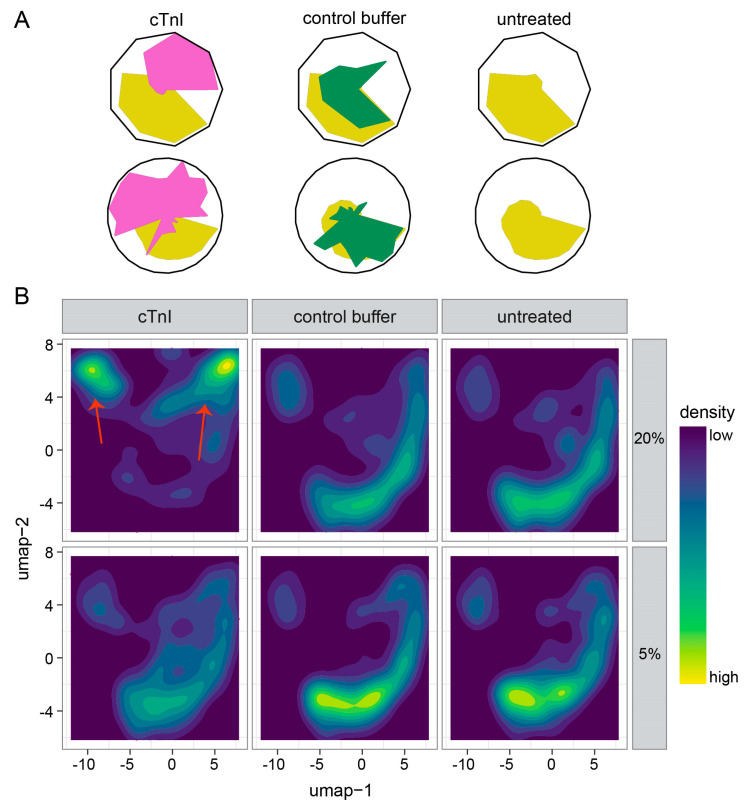
Assessment of multiparametric morphological phenotypes. cTnI plasma induced a shift in cardiomyocyte morphology in averaged and single-cell resolutions. (**A**) Multiparametric morphological phenotypes after incubation with 5% and 20% plasma concentration of cTnI or control buffer and untreated mouse plasma. The average phenotype is visualized utilizing radar plots with multiple feature scales ordered radially to display a multitude of feature values in one figure. The feature values are plotted on the feature scales. For visualization, the inner area of the plots is filled with the following colors: yellow for the untreated control, green for the control buffer, and pink for cTnI. (**B**) Analysis of cTnI single-cell morphology displays the heterogeneity of cardiomyocytes. A UMAP representation of the morphological changes on a single-cell level is shown. The UMAP plots highlight the different phenotypes between cTnI 20% compared to the controls (**upper row**); red arrows highlight the cTnI-induced specific phenotypes. At 5% plasma concentration, this cTnI-induced phenotype is only weakly induced (**lower row**).

## Data Availability

The datasets generated and analyzed during the current study are available from the corresponding author upon reasonable request.

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
