# Peer review of "Cardiac Troponin I Antibodies Induce Cardiomyocyte Damage and Alter Cell Morphology"

_ijms, 2025, doi:10.3390/ijms262010005_

Round 1

Reviewer 1 Report

Comments and Suggestions for Authors

The topic is innovative, with great clinical significance, since the measurement of anti cTnI abs may be useful as a dieagnostic criteria in ptients with autoimmune myocarditis. Overall, the article is well-written, and the presentation of the results, graphs, images are impecable. However, several issues should be addressed in order to improve the article for scientific and general audience:

  1. It is recomended to comment a bit more about EAM model in the Introduction section, its advantages, how it mimics autoimmune myocarditis in humans.
  2. Additionally, please describe the goal of the study more specifically in the end of Introduction section.
  3. Please, give the rationale for using female mice and age of mice, since many studies using EAM model propose using males , 6-8 weeks old animals and s.c application instead of intraplantary into footpad, like usually described in the literature?. Additionally give more details about immunization of animals. 
  4. For ELISA technique it is not necessary to describe detailed procedure, as it is familiar to many researchers, instead write `according to manufacturer`s instructions` and give exact ELISA kit you used with manufacturer and LOT number.
  5. If possible include another echocardiographic parameter regarding LV dimensions, such as LVPW, IVS, LVID, since it could be useful to assess the possible development of dilative cardiomyopathy characteristic for EAM in chronic phase. What formula did you use to calculate EF and FS?
  6. Discussion section should be improving by adding more studies on EAM model to compare with.

Reviewer 2 Report

Comments and Suggestions for Authors

The manuscript by Furkel, Zirkenbach et al. dives into the role of cardiac-specific autoantibodies, particularly anti-cardiac troponin I (cTnI) antibodies, in causing myocardial damage and remodeling. The authors utilize both a mouse model of experimental autoimmune myocarditis and in vitro assays with neonatal rat cardiomyocytes (NRCMs). This topic is timely and significant, especially given that the potential harm of cardiac autoantibodies in heart failure is still a crucial and unresolved issue. By combining in vivo and in vitro experiments with morphological profiling, the authors present an interesting dataset. The structure of the manuscript is sound, and there are some important findings; however, a few areas need clarification and expansion to strengthen the overall conclusions.

Major comments:

- The study confirms that anti-cTnI antibodies are pathogenic, consistent with prior reports (Okazaki et al., Wu et al., Furusawa et al.). While this replication is valuable, the novel mechanistic contribution is not entirely clear. The authors should better highlight what is new compared to previous work.

- Additional mechanistic experiments (e.g., blocking Fc receptors, testing Fab fragments, or signaling pathway inhibition) would help establish whether the observed effects are antibody-mediated signaling vs. nonspecific plasma effects. Plasma is a complex mixture of proteins and cytokines, not just antibodies. The authors should confirm that the effects are specifically due to anti-cTnI aAbs (for example, using antibody depletion or comparison with isotype controls).

- Were plasma samples heat-inactivated to eliminate complement? Complement activity could independently cause cell damage.

- In Figure 1, the mice weights at different time points (baseline, day 14) should be included. In addition, heart weight normalized to femur length would be an important parameter to include.

- Could the authors please elaborate on how the inflammation score shown in Figure 2A was calculated?

- In Figure 3, in addition to ejection fraction, the authors should also provide other cardiac parameters (e.g., cardiac output, stroke volume). The figure caption should indicate the number of mice used for this analysis.

- Please clarify whether the number of mice used in the experiments is sufficient to achieve statistical power, given that some graphs display considerable variability with high standard deviations.

- Figure 4 G is difficult to visualize. Please provide a higher-magnification images and include single-channel images in addition to the merged image.

- The C-MORE analysis is promising, but the data presentation should be improved. Figures should clearly illustrate which parameters are most affected (cell size, shape, texture). Are these changes reproducible across biological replicates?

Minor comments:

-Please check the text carefully (line 203 ... Desmin von cytoplasm) (line 337-338... Error! Reference source not found.A).

-Can you please better explain lines 361-362: “Therefore, the area infiltrated by immune cells or fibrosis was considered in relation to the whole heart section by eye and was specified in percentage.”

Round 2

Reviewer 1 Report

Comments and Suggestions for Authors

Authors have successcully answered almost all suggestions of the reviewer, and the article is now much improved and suitable for publication.

Reviewer 2 Report

Comments and Suggestions for Authors

I appreciate the authors’ efforts in answering and addressing my concerns, and I endorse the publication of this article.